# A decade of glaciological and meteorological observations in the Arctic (Werenskioldbreen, Svalbard)

Dariusz Ignatiuk[1], Małgorzata Błaszczyk[1], Tomasz Budzik[1], Mariusz Grabiec[1], Jacek A. Jania[1], Marta Kondracka[1], Michał Laska[1], Łukasz Małarzewski[1], Łukasz Stachnik[2]

[1]University of Silesia in Katowice, Katowice, 40-007, Poland
[2]University of Wrocław, Wrocław, 50-137, Poland

*Correspondence to*: Dariusz Ignatiuk (dariusz.ignatiuk@us.edu.pl)

**Abstract.** The warming of the Arctic climate is well documented, but the mechanisms of Arctic amplification are still not fully understood. Thus, monitoring of glaciological and meteorological variables and the environmental response to accelerated climate warming must be continued and developed in Svalbard. Long-term meteorological observations carried out in situ on glaciers in conjunction with glaciological monitoring are rare in the Arctic and significantly expand our knowledge about processes in the polar environment. This study presents glaciological and meteorological data collected in 2009-2020 in southern Spitsbergen (Werenskioldbreen). The meteorological data are comprised of air temperature, relative humidity, wind speed, shortwave and longwave upwelling and downwelling radiation on 10 minutes, hourly and daily resolution (2009-2020). The snow dataset includes 49 data records from 2009-2019 with the snow depth, snow bulk density and SWE (snow water equivalent) data. The glaciological data consist of seasonal and annual surface mass balance measurements (point and glacier-wide) for 2009-2020. The paper also includes modelling of the daily glacier surface ablation (2009-2020) based on the presented data. The datasets are expected to serve as local forcing data in hydrological and glaciological models and validation of calibration of remote sensing products. The datasets are available from the Polish Polar Database (https://ppdb.us.edu.pl/) and Zenodo (https://doi.org/10.5281/zenodo.6528321, Ignatiuk, 2021a; https://doi.org/10.5281/zenodo.5792168, Ignatiuk, 2021b).

## 1 Introduction

Long-term meteorological observations carried out in situ on glaciers in conjunction with glaciological monitoring are rare in the Arctic and may be used to expand knowledge about processes in the polar environment. Terrestrial meteorological monitoring alone does not always adequately address the needs of numerical modelling as well as validation and calibration of satellite products regarding glaciers (Pelliccotti et al., 2014; Gabbi et al., 2017). The warming of the Arctic climate is well documented, but the mechanisms of Arctic amplification are still not fully understood (IPCC, 2019). Both, climate and ocean variables have fluctuated in Svalbard in the last decades (Nordli et al., 2014; Schuler et al., 2014; Isaksen et al., 2016;

Vikhamar-Schuler et al., 2016; Walczowski et al., 2017; Osuch and Wawrzyniak, 2017; Førland et al., 2020; Wawrzyniak and Osuch, 2020), which causes progressive and ongoing changes in the cryosphere (Błaszczyk et al., 2013; Wawrzyniak et al., 2016; Box et al., 2018; Grabiec et al., 2018; Nuth et al., 2019; van Pelt et al., 2019; Schuler et al., 2020; Błaszczyk et al. 2021;). According to the data published in by SIOS data access portal (https://sios-svalbard.org/) and Meteorological bulletin Spitsbergen-Hornsund (https://hornsund.igf.edu.pl/weather/) 2020 was the year with the warmest summer in the history of instrumental observations in Svalbard (the mean JJA- June/July/August temperature was 7.2°C, about 3°C above the climatological normal at the Svalbard Airport meteorological station). In Hornsund the same summer months mean was 4.8°C (only 1.2°C higher than the local normal). The highest air temperature since the beginning of measurements was recorded on July 25th, 2020: 21.7°C and 16.5°C at Svalbard Airport and the Polish Polar Station in Hornsund, respectively. Moreover, in 2019 the sea ice area on the Arctic Ocean reached the second minimum extent in the history of satellite measurements since 1979 (Yadav et al., 2020). While the summer of 2021 was colder and the minimal Arctic sea ice extent significantly larger, acceleration of the climate warming trend is proved despite interannual variations (Hanssen-Bauer et al., 2019). Such acceleration causes significant changes in the cryosphere of Svalbard and is particularly reflected in the faster melting of glaciers and thawing of the permafrost (Schuler et al., 2020, Christiansen et al., 2021). It also stimulates faster energy and mass exchange between the atmosphere, cryosphere and ocean. The above examples of transition in air temperature, sea ice extent or glacier and permafrost melting demonstrate regional differences in climate warming and subsequently response of other environmental components. Therefore, monitoring of such parameters and the environmental response to climate change is recommended to be carried out in Svalbard, where climate warming is one of the most dynamic (Nordli et al., 2014; Isaksen et al., 2016). Long-term observations allow for better quantification of observed changes and facilitate their more profound understanding. This study presents the unique Arctic glaciological and meteorological data collected in 2009-2020 in southern Spitsbergen.

## 2 Study Area

Werenskioldbreen is a well-studied, polythermal glacier located in South Spitsbergen (Figure 1)) (Baranowski, 1982; Pälli et al., 2003; Grabiec et al.., 2012; Ignatiuk et al., 2014; Stachnik et al., 2016a; Stachnik et al., 2016b; Sułowicz et al., 2020). This valley-type glacier covered an area of 27.1 km$^2$ in 2008 (Ignatiuk et al., 2014) and 25.7 km$^2$ in 2017 (current study) in a catchment area of 44 km$^2$. Werenskioldbreen is divided by a medial moraine into south-eastern part Slyngfjellbreen and the northern part Skilryggbreen accumulation area (Figure 1). The glacier's forefield is closed by a distinct arc of the ice-cored terminal moraine with one river gorge. Such a hydrological system allows the glacier basin to be treated as a well-defined research laboratory for many hydrological and interdisciplinary studies (Majchrowska et al., 2015; Stachnik et al., 2016b; Łepkowska and Stachnik, 2018; Gwizdała et al., 2018; Stachnik et al., 2019; Osuch et al., 2022). The glacier is situated 15 km to the north from the Polish Polar Station Hornsund. The Stanisław Baranowski Spitsbergen Polar Station (University of

Wrocław), a small field station is located at the southern edge of the Werenskioldbreen terminal moraine. Both facilities greatly simplify the accessibility and logistics of research and monitoring projects.

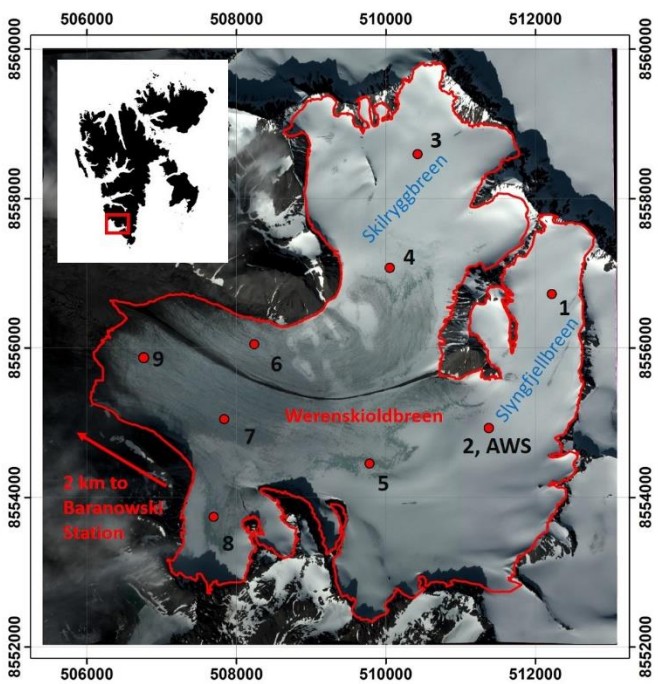

**Figure 1: Location of mass-balance stakes (1-9) in 2009-2020 and the automatic weather station (AWS) on Werenskioldbreen (background: GeoEye, 2010/08/10)**

**3 Instruments and methodology:**

**3.1 Meteorological monitoring**

The automatic weather station (AWS) is located at an altitude of 380 m above sea level (Figure 1), close to the average equilibrium line altitude (ELA) for the years 1959 – 2008 (Noël et al, 2020). The station was installed on the glacier on 15 April 2010. The AWS was mounted on a long steel mast placed in the ice drilling hole (ca. 6 m deep). In the following years,

as ablation progressed, the sensors were lowered or the mast was replaced with a new one in close proximity to the original location. The recording of variables (air temperature, humidity, wind speed, shortwave and longwave radiation) has started on 17 April 2010 (Table 1). The Kipp & Zonnen CNR4 consists of two CM3 pyranometers, two CG3 pyrgeometers and temperature sensors (PT100). Pyranometers (180° solid angle) have a glass dome and measure radiation in the range from 300 to 2 800 nm. One of the pyranometers directed upwards measures downwelling radiation, and the second one directed

downwards measures solar radiation reflected from the earth's surface (upwelling radiations). Pyrgeometers (180°-150° solid angle), has silicone windows, which allow radiation measurements in the range from 4500 nm to 42000 nm. Like the

pyranometers, the pyrgeometers point in opposite directions (upwards and downwards). One of the pyrgeometers measures the long-wave radiation coming from the atmosphere, the second one the long-wave radiation from the ground surface. In 2016, the A100R cup anemometer was replaced with the Gill WindSonic sensor, which allowed for measuring wind speed. Damaged sensors were replaced during the spring or autumn service to maintain data continuity.

Table 1. Automatic weather station (AWS) sensors specification (Werenskioldbreen, Svalbard). All sensors were set in the spring so that they were about 1.5 meters above the snow surface, height has been varied during the season.

| Variable | Sensor/producer | Operating range | Accuracy | Period of operation (percent of gaps in dataset) |
|---|---|---|---|---|
| Air temperature | 107/Campbell Scientific | -35 … + 50°C | ± 0.2 °C (over range 0 … 50°C) | 04/2010 – 05/2016 (6%) |
| Air temperature | HMP155 (PT100)/Vaisala | -80 … + 60°C | ± Accuracy at −80 … +20 °C  (0.226 - 0.0028 × temperature) °C | 05/2016 – 12/2019 (60%) |
| Relative humidity | HOBO / ONSET COMPUTER | 0 … 100% RH | ±2.5%  (10% – 90% RH) ; ±5% (below 10% and above 90%) | 03/2011 - 09/2011 (0%) |
| Relative humidity | HMP45AC / Vaisala | 0 … 100% | ± 2% (0 – 90% RH) ± 3% (90 – 100% RH) | 05/2016 – 08/2019 (55%) |
| Wind speed | A100R/Vector Instruments | 0 … 75 m s$^{-1}$ | ± 0.1 m/s (0.3 – 10 m s$^{-1}$); ± 1% (10 – 55 m s$^{-1}$); ± 2% (> 55 m s$^{-1}$) | 09/2010 – 05/2016 (14%) |
| Radiation<br>-    shortwave: downwelling and upwelling<br>-    longwave: downwelling and upwelling | CNR4 /Kipp&Zonen | Shortwave: 300 – 2 800 nm Longwave: 4 500 – 42 000 nm | Pyranometer: Uncertainty in daily total < 5% Pyrgeometer: Uncertainty in daily total < 10% ± 6% (-40 – 80°C) ± 25 W m$^{-2}$ at 1 000 W m$^{-2}$ | 09/2010 – 05/2016 (1%) |
| Datalogger | CR1000/Campbell Scientific | -40 – 50°C | | 04/2010 – 05/2020 (-) |
| Ablation and accumulation | SR50/Campbell Scientific | 0.5 – 10 m | ± 1 cm or 0.4% | 9/2010 – 12/2019 (-) |

In the 2010/2011 season, measurement recording t by the logger was performed every 1 min. Due to high energy demand during the polar night, the sampling time was changed to an instantaneous measurement every 10 minutes. Calibration and

testing of sensors were performed regularly during spring expeditions based on the infrastructure of the Polish Polar Station Hornsund.

## 3.2 Glaciological monitoring

In 2009-2010, nine mass-balance stakes were installed on the Werenskioldbreen. The locations have been chosen to cover the elevation range from 117 m a.s.l. to 515 m a.s.l to create altitudinal profiles along the northern and southern tributaries of the glacier (Figure 1).

 The stakes, 6 to 8 meters long, were embedded in the glacier by a steam drilling rig or by Kovacs Ice Coring System (ICS). The mass-balance stakes were measured twice a year (spring-autumn, 2009-2013) during the winter maximum accumulation (April-May) and at the end of the ablation season (September-October) or once a year (at spring, since 2014). The measure of winter accumulation was determined during the spring campaigns. The properties of snow cover (bulk snow density, snow depth, SWE) were measured in snow pits (a 100 cm$^3$ snow gauge by Winter Engineering was used to determine the snow density of subsequent layers) or shallow core boreholes (ICS). During the measurements, repeated soundings of the snow depth were also performed with avalanche probes. In the absence of the autumn campaign, boreholes have been drilled near each stake in order to accurately determine the amount of summer ablation and possible summer accumulation.  Measurements during the autumn campaign did not always take place after or at the end of the ablation season. This was due to the logistics of the expeditions and the extension of the ablation season. In the case of availability of data from the SR50A sensor, ablation or accumulation corrections were also made if the winter or summer season ended later than the date of field observations. Some of the ablation stakes have been damaged every few years. They have been broken by wind, polar bears, melt out from the ice or been buried by snow. The network of ablation stakes was supplemented and renovated during maintenance visits. Unfortunately, recent years have resulted in large gaps in measurements due to the pandemic travel restrictions (years 2020 and 2021). Detailed information on the temporal availability of glaciological data is presented in Table 2.

Table 2. Overview of mass balance and snow cover measurements on ablation stakes and infrastructure maintenance in years 2009-2020 on Werenskioldbreen (Svalbard), where: S – spring campaign (winter balance, April-May), A – autumn campaign (summer balance, August-September, can be performed next year spring), X – lack of stake, SP – snow pit (SWE data), KD – ICS drilling (SWE data).

| Stake no. | 1 | 2 (+AWS) | 3 | 4 | 5 | 6 | 7 | 8 | 9 |
|---|---|---|---|---|---|---|---|---|---|
| Coordinates (UTM 33N), height (geoid EGM_96) | N8556724; E512219; 515 | N8554930; E510423; 384 | N8558594; E510423; 471 | N8557076; E5100481; 392 | N8554448; E509786; 308 | N8556047; E508243; 188 | N8555045; E507837; 199 | N8553738; E507697; 277 | N8555956; E506449; 120 |
| 2009 | S, A, SP, | S, A, SP, | S, A, SP, | S, A, SP, | S, A, SP, | S, A, SP, | S, A, SP, | S, A, SP, | X |
| 2010 | S, A, SP, | S, A, | S, A, | S, A, | S, A, SP, | S, A, | S, A, SP, | S, A, | S, A, SP, |
| 2011 | S, A, SP, | S, A, | S, A, SP, | S, A, SP, | S, A, | S, A, | S, A, SP, | S, A, SP, | S, A, |

| Year | | | | | | | | | |
|------|---|---|---|---|---|---|---|---|---|
| 2012 | S, A, | S, A, SP, | S-X, A | S-X, A | S-X, A | S, A | S, A, SP | S, A | S, A |
| 2013 | S-X, A-X, | X, | S,KD,A | S, KD, A | S-X, A-X, | S, KD, A | S, SP, | S, KD, | S-X, A |
| 2014 | S, A, SP, | X, | S, A | S, A | S, A-X, SP | S, A | S, A, SP, | S, A | S, A-X |
| 2015 | S, A-X, SP, | X, | S, A | S, A | S, A, SP | S, A | S, A | S, A-X | S, A-X, SP |
| 2016 | S, A-X, KD, | X, KD | S, A | S, A | S-X, A-X, | S, A-X | S, A | S, A | S-X, A-X |
| 2017 | S, A, | X, KD, | S, A, KD | S, A, KD | S, A, KD | X, A | S, A, KD, | S, A | S, A-X, KD, |
| 2018 | S-X, A, KD, | X, KD, | S, A, KD | S, A, KD | S, A | X, | S, A, KD, | X, | S, A, KD, |
| 2019 | X, KD | X, KD | S-X, | S, A | S, KD, A | X, | X, KD, | X, | S, A, KD, |
| 2020 | S | X | S | S | S | X | S, | X | S, |

Based on the data collected, the following glaciological variables are available: seasonal and annual point and glacier-wide surface mass balance, snow depth, bulk snow density and SWE at the measuring points.

The analyses of the glacier's surface mass balance excepted field measurements were based on altitude zones determined from digital elevation models (DEM). Two DEMs with geoidal height (EGM2008) were used, one generated from SPOT image acquired on 1 September 2008 (Ignatiuk et al., 2014) for years 2008 – 2019 and Pleiades high-resolution images taken on 20 August 2017 (Błaszczyk et al., 2019) for year 2020.

## 4 Meteorological observations

### 4.1 Air temperature and radiation

The air temperature data forms the most homogeneous series for 2010-2016 (Figure 2a). In 2017-2020 the data gaps were already significant due to a series of failures of the instruments. Also, for 2010-2016, net radiation balance data (short- and long-wave radiation) are available (Figure 2b,c). In 2017-2020, radiation data included only downwelling shortwave radiation.

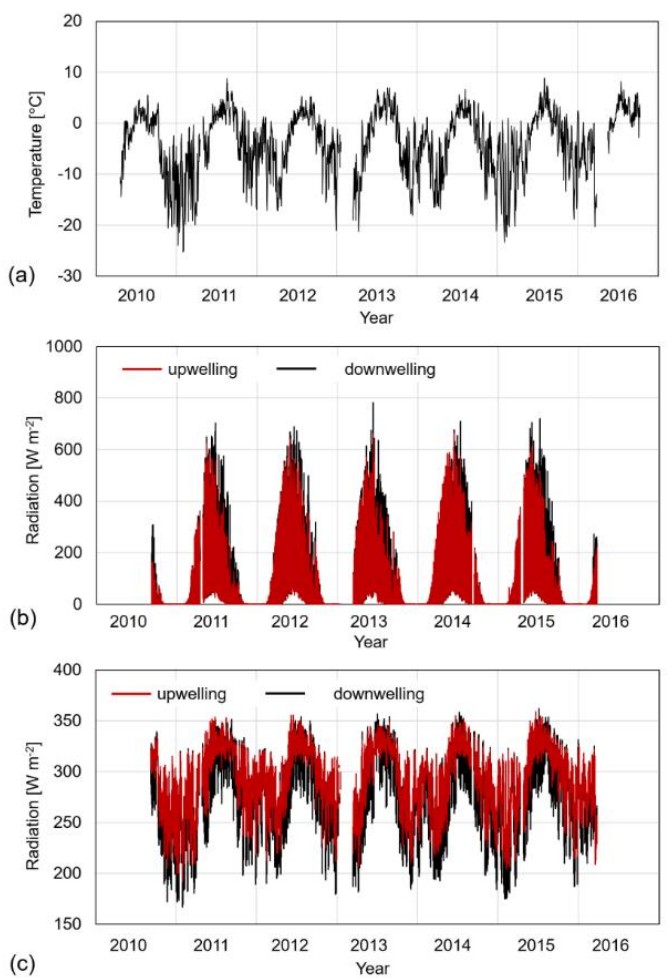

Figure 2: Time series of meteorological variables from 2010 to 2016 on Werenskioldbreen, including daily average air temperature (a), hourly average shortwave radiation (b) and hourly average longwave radiation (c) for downwelling (black line) and upwelling (red line) radiation.

For years with full data continuity (Figure 2a), air temperature monthly and yearly averages were calculated and then compared with the data from the Polish Polar Station Hornsund (Wawrzyniak and Osuch, 2020). The average difference in the annual temperature (2011, 2012, 2014, 2015) between glacier (380 m a.s.l.) and the Polish Polar Station Hornsund (8 m a.s.l.) was -2.7°C , which gives an average temperature lapse rate 0.72/100 m (annual values varies from 0.55 to 0.80). We have calculated the significance of the trend presented in this study using the non-parametric modified Mann – Kendall test (Hamed and Rao, 1998) with considering the effect of autocorrelation of time series. The slope of the trend was calculated using Sen's method (Sen, 1968). The test indicated the statistically significant increasing temperature trend in the period 2010-2016 (with the significance level alfa = 0.05 and p-value = 0.036) taking into account the 12-month seasonality the Sen's slope was 0.02. Gaps in the data were filled based on the relations between air temperature measured on the PPS in Hornsund and air

temperature on the WRN glacier ($R^2 = 0.96$). According to the Wawrzyniak & Osuch (2019) estimated slope of trend for air temperature between 1979–2018 at the Hornsund station was estimated as 1.14° C per decade. Increasing distance between the glacier surface and the sensors during the season is affecting on the air temperature measurements. Periodic measurements of the vertical temperature gradient between 0.5 and 4 m carried out at the AWS indicate that the air temperature in the atmospheric boundary layer changes by about 0.2°C per 1 m during the ablation season. Downwelling shortwave radiation reaches its maximum during the middle of the polar day (June). Its annual course is governed by the occurrence of polar day and night, while its daily course is governed the height of the sun above the horizon. The reflected shortwave radiation (upwelling) is a function of the surface albedo. In the spring and early summer, we observed the highest values of reflected radiation due to the presence of snow cover on the glacier. In the further part of the ablation season (July-August), we noticed a sudden decrease in reflected radiation (Figure 2b) as a result of the disappearance of snow cover at the measurement site (AWS) and the appearance of glacial ice on the surface. The decrease in upwelling shortwave radiation can be slight (e.g. 2012) when the melting of snow cover occurs mainly as a result of surface ablation or abrupt (e.g. 2015) when significant rainfall led to a sudden change in the albedo on the glacier. The maximum values of downwelling/upwelling longwave radiation (Figure 2c) usually occurred in summer and autumn. The values in winter and spring are lower, which in general shows similar patterns with the seasonal variations in air temperature. The values above 316 W m$^{-2}$ of outgoing longwave radiation may be caused by the presence of the water under the station or the presence of sediment or cryoconite. Both of these situation were observed at AWS. In the he view of the CNR4 sensor (180o), there is also a mast with a logger, sensors and a solar panel, what cause distorts of the observations. These problems are difficult to eliminate. Assuming the homogeneity of the surface around AWS, increasing the distance of the CNR4 sensor from the glacier surface should not affects its measurements.

### 4.2 Other variables

The AWS measured relative humidity, wind speed, ablation and accumulation of snow (for the time span see Table 1). These sensors were installed at the station depending on the needs of the ongoing projects. Not all of them could be connected to the datalogger at the same time. Servicing only once a year, causing a higher failure rate for these sensors. Therefore, the data obtained for these variables are not continuous and not homogenous for the entire observation period. Nevertheless, these data are available and are of great value for solving specific scientific problems like rain on snow events (Łupikasza et al., 2019) or supplying data to other models (Dacaux et al., 2019).

### 5 Glaciological observations

### 5.1 Point ablation and accumulation

Measurements on mass-balance stakes (Figure 1, Table 2) were performed in accordance with the recommendations and guidelines contained in the Glossary of Glacier Mass Balance and Related Terms (Cogley et al., 2011). After Cogley et al.

(2011) it was assumed that accumulation is always positive, while ablation is negative. Therefore, the calculation of the point mass balance is Eq. (1):

$$b_a = c_a + a_a = b_w + b_s = c_w + a_w + c_s + a_s ,$$  (1)

where: ba – annual balance at a point, ca – annual accumulation, aa – annual ablation, bw – winter balance, bs – summer balance, cw – winter accumulation, aw – winter ablation, cs – summer accumulation, as – summer ablation

A method of determining point mass balance on the glacier surface includes measurements at stakes and in snow pits or boreholes. The high of each stake above snow/ice is measured twice in the maximum of winter accumulation and at the end of the summer ablation. The measurements also include depth probing and density sampling of the snow and firn (see section

5.2). They are made at single points, the results from a number of points being extrapolated and integrated to yield the surface mass balance over the entire glacier (Cogley et al., 2011; see section 5.3). The error of point mass balance was estimated using the total differential function using general equation:

$$\Delta f(x_1, x_2, \dots x_n) = \sqrt{\left(\frac{\partial f}{\partial x_1}\right)^2 (\Delta x_1)^2 + \left(\frac{\partial f}{\partial x_2}\right)^2 (\Delta x_2)^2 + \cdots + \left(\frac{\partial f}{\partial x_n}\right)^2 (\Delta x_n)^2}$$  (2)

Where: Δf, Δx – error of the variable, ∂f/∂xn - partial derivative, $x_1, x_2, \dots x_n$ – variables.

Base on the eq. (2) we've calculated the error of winter, summer and annual point surface mass balance using equations:

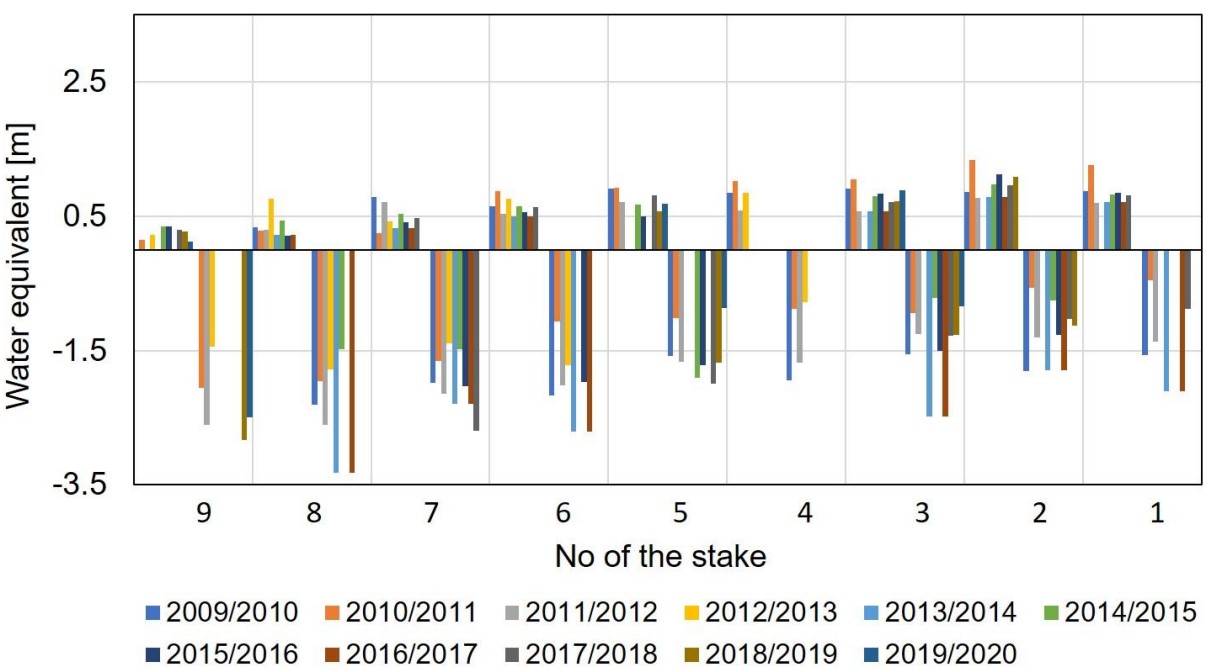

**Figure 3: Winter and summer balance at the point on 1-9 mass-balance stakes in years 2009-2020 For the location/elevation of the stake see Figure 1 and Table 2. Stakes have been lined up according to a height above sea level.**

The dataset includes point winter and summer mass balance measurements on mass-balance stakes in 2009-2020 and the calculated point annual mass balance. The data allow the analysis of the spatial and seasonal variability of accumulation and ablation at points on the glaciers at different altitudes. The analysis of the winter balance (Figure 3) shows the interannual fluctuations in snow accumulation in the entire altitude profile of the glacier. The analysis of point winter balance shows the smallest interannual fluctuations on the glacier snout (stake 9) and in the sheltered upper glacier cirque (stake 8). In the case of the point summer balance, the greatest interannual changes are observed in the middle zone of the glacier (200 – 400 m a.s.l.). This is due to a longer ablation period and higher temperatures not previously recorded at these altitudes. The variability of the annual surface mass balance is dominated by the summer surface mass balance (Østby et al., 2011; Grabiec et al., 2012; Van Pelt et al., 2019).

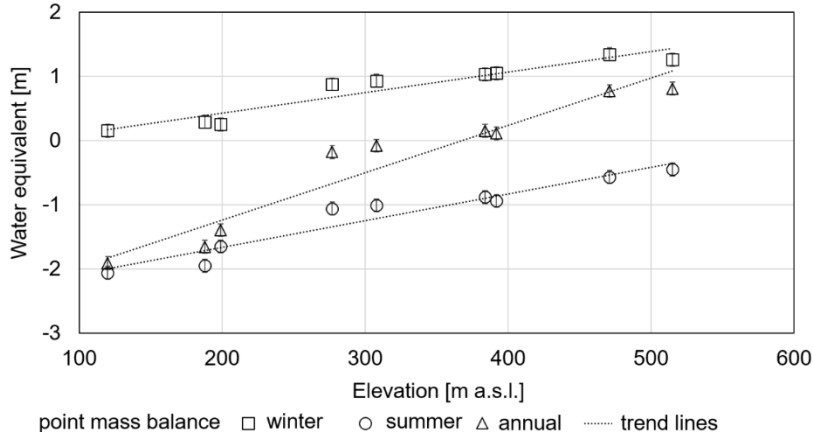

**Figure 4: Examples of winter, summer and annual point mass balance on Werenskioldbreen (season 2010/2011). Whiskers show an error (total differential function).**

Each of the balance years can be considered separately (Figure 4). Winter accumulation in the analysed period was generally low. The last significant accumulation on Werenskioldbreen took place in 2011. A slight accumulation in the highest parts of the glacier was also observed in 2013, 2015 and 2020. Observations from 2020, however, may be biased by unqualified substitutive observers due to the pandemic situation. In all other years, the ELA was above the highest monitored stake, no 1 (Table 2). The data on point mass balance components are crucial for calculations of the glacier-wide surface mass balance. These data have very high importance for different modelling purposes, e.g. hydrology and glacier drainage modelling, total water discharge from the glacier, sea level rise models and validation of remote sensing products.

## 5.2 Snow Water Equivalent (SWE)

In the years 2009-2019, 49 samplings (shallow drilling or snowpits) were made on the glacier during the spring measurement campaigns in order to determine the bulk snow density, and thus SWE. In the case of snowpit measurements, the density was measured for each homogeneous layer. The bulk snow density for the snow profile was then averaged weighted by layer thickness.

Whereas the bulk snow density during the drilling of snow cores was calculated based on the length and weight of each core in the profile. SWE is calculated based on the following equation (Sturm et al. 2010):

$$SWE = h_s \frac{\rho_s}{\rho_w} \tag{3}$$

$$\Delta SWE = SWE \sqrt{\left(\frac{\Delta h_s}{h_s}\right)^2 + \left(\frac{\Delta \rho_s}{\rho_s}\right)^2} \tag{4}$$

Where: $h_s$ – snow depth [m] , $\rho_s$ – bulk snow density [kg m$^{-3}$] , $\rho_w$ –density of water [1kg m$^{-3}$], $\Delta h_s$ – error of the snow depth [0.01 m], $\Delta \rho_s$ – error of the bulk snow density [10 kg/m$^3$].

The average density of snow cover ranges from 386 to 447 kg m$^{-3}$ (Table 3). The highest snow density values were noted in 2012. They are related to the extremely warm conditions in the winter season 2011/2012 with the heavy rainfall (Łupikasza et al., 2019) during the winter and caused probably by the inflow of warm Atlantic water (the fjords of south-west Spitsbergen did not freeze).

Table 3. Average snow depth and bulk snow density based upon data from sampling points (snow pits and drilling cores) on Werenskioldbreen in years 2009-2019.

| Year | 2009 | 2010 | 2011 | 2012 | 2013 | 2014 | 2015 | 2016 | 2017 | 2018 | 2019 |
|---|---|---|---|---|---|---|---|---|---|---|---|
| Number of sampling | 8 | 3 | 5 | 2 | 6 | 3 | 3 | 2 | 6 | 4 | 7 |
| Average snow depth [m] | 1.79 | 1.90 | 1.65 | 1.35 | 1.06 | 1.73 | 1.57 | 1.90 | 1.64 | 1.53 | 1.60 |
| Average bulk snow density [kg/m$^3$] | 434 | 415 | 412 | 447 | 386 | 391 | 387 | 407 | 427 | 419 | 410 |
| Avarange SWE [m w.e.] | 0.77 | 0.81 | 0.68 | 0.61 | 0.52 | 0.68 | 0.61 | 0.79 | 0.71 | 0.64 | 0.65 |

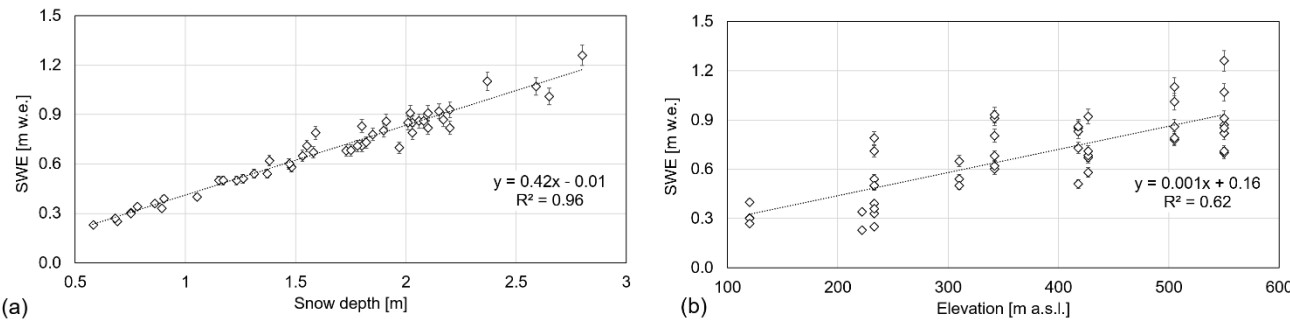

**Figure 5: Relationship between SWE and snow depth (a) and SWE and elevation m a.s.l. (b). Whiskers show an error (total differential function, eq.4).**

SWE values show a very high correlation with the snow depth ($R^2 = 0.96$, Figure 5a) and lower correlation with the altitude above sea level ($R^2 = 0.62$, Figure 5b). Uszczyk et al. (2019) found the relationship between the bulk snow density and the snow depth on Hansbreen located next to Werenskioldbreen. It was observed that the bulk snow density increases with snow depth. The long-term data collected on Werenskioldbreen has not confirmed this correlation. In fact, in some seasons it is the opposite, i.e. thinner snow cover in the lower zones of the glacier has the highest bulk snow density. Seasonal variability can

be explained by various meteorological conditions during the accumulation season. The differences between Werenskioldbreen and Hansbreen can most likely be explained by different orographic conditions and exposure, which affects snow blowing and snow deposition.

**5.3 Glacier-wide surface mass balance**

While the mass balance is measured on many glaciers, the data series rarely exceeds 10 years (Schuler et al., 2020). Multi-

235     year data series, such as those from Werenskioldbreen, represent a unique value for tracking long-term changes in the Arctic environment. Calculation of the mass balance was based on point winter and summer balance analyses and digital elevation models. The point measurements are extrapolated over the glacier surface determining the balance as a function of altitude and averaging them, using the weights determined from the distribution of the glacier surface as a function of altitude (Cogley et al., 2011). The error was estimated using the total differential function.

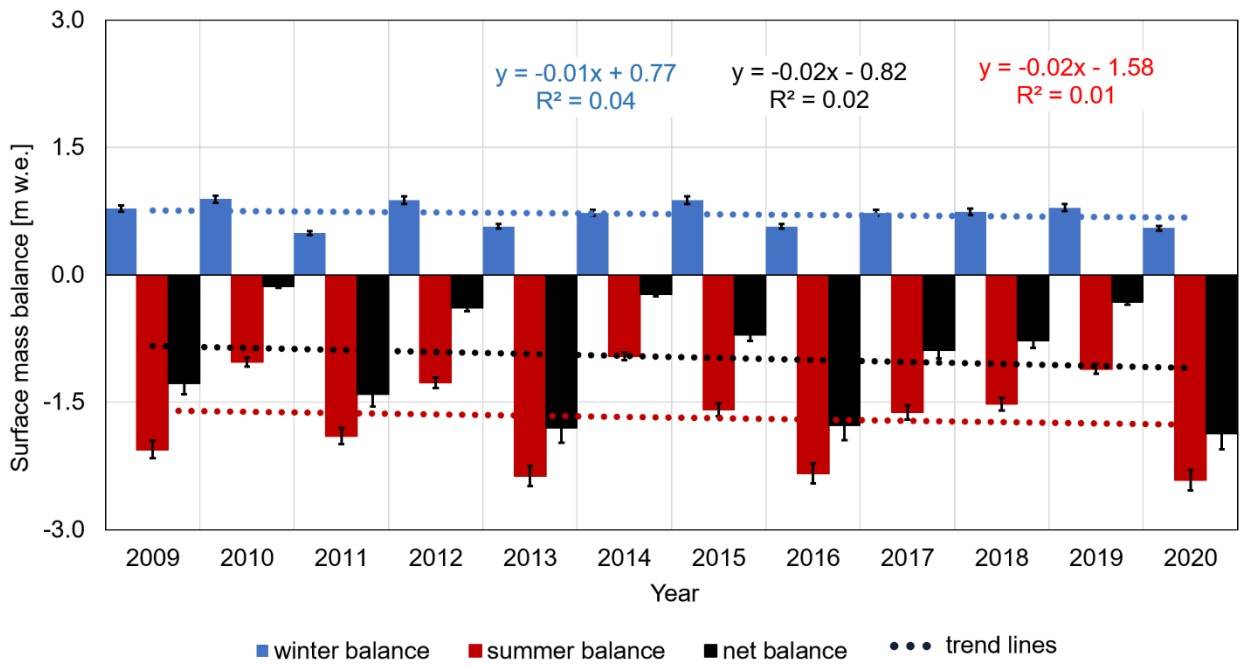

**Figure 6: Annual surface mass balance and its components of Werenskioldbreen in 2009-2020. Blue bars - winter mass balance, red bars - summer mass balance, green bars - net mass balance. The results for 2019 may be understated (field measurements performed by the non-expert crew).**

The largest fluctuations are observed in the summer balance, which depends on the interannual changes in the duration of the positive air temperatures and thus the length of the ablation season. The winter balance shows greater stability, however, over the decade, the amount of snow accumulation is downward. This entails a negative trend in the surface mass balance of Werenskioldbreen (Figure 6). Based on the trend lines, it can be concluded that acceleration of mass loss decreases by 0.09 m w.e. per decade, while the summer balance decreases by -0.14 m w.e. for a decade. This gives us an acceleration of the ablation by 0.23 m w.e. for a decade on Werenskioldbreen.

The significance of the trends using the non-parametric modified Mann – Kendall test (Hamed and Rao, 1998) showed there is no statistically significant trend in 2009-2020 ($\alpha = 0.05$). The Sen's slope was -0.01, -0.12 and -0.02 for winter, summer and annual glacier-wide mass balance respectively. Hagen et al. (2012) shown that it is impossible to give any trend for the glacier mass balance data for so short time period.

Grabiec et al. (2012) have used monthly values of air temperature and precipitation from the meteorological station at Hornsund and the reanalysis ERA-40 data to hindcast the mass balance of Werenskioldbreen for the years 1959-2002. The average correlation coefficient of the modelled and observed mass balance in the 5 seasons (1994, 1999–2002) was 0.67 (meteorological data) and 0.70 (ERA-40). The average glacier-wide winter surface mass balance for 1959-2002, according to

the model, was 0.81 m water equivalent (w.e.) (ERA-40 data) and 0.87 m w.e. (meteorological data) which, compared to the last decade (average winter glacier-wide surface mass balance 0.72 m w.e.), decreased by 7 and 13% respectively. The glacier-wide summer surface mass balance decreased from -1.23 m w.e. in 1959-2002 to -1.68 m w.e. in 2009-2020 (37%) in comparison to the meteorological model and from -1.14 m w.e. in 1959-2002 (47%) for the ERA-40 data model. Detailed analysis of glacier-wide summer surface mass balance data from modelling (1979-2005) and observations (2009-2020) shows an increase in the average 10-year glacier-wide summer surface mass balance from -1.16 m w.e. in 1979-1988, through -1.35 and -1.55 in 1989-1998 and 1999-2005, respectively, to -1.68 in 2009-2020. A natural consequence of increasing the glacier-wide summer surface mass balance is also a much more negative average annual glacier-wide surface mass balance in the last decade (-0.97 m w.e.) compared to the years 1959-2002 (-0.35 m w.e. for the meteorological data model and -0.34 m w.e. for ERA-40 data model).

## 5.4 Daily surface ablation

The influence of air temperature on the glacier surface ablation has been the subject of numerous studies. The coefficient of determination between the annual ablation and the sum of positive daily air temperature was calculated as 0.96 ($R^2$) by Braithwaite and Olsen (1989). High correlation is caused by the strong dependence between the air temperature and the components of the energy balance (Hock, 2003). Ohmura (2001) presented the physical basis for the application of temperature ablation models, the relationship between air temperature and long-wave radiation of the atmosphere, sensible heat and incident short-wave radiation. The basic temperature ablation model is given by the equation (Braithwaite, 1995):

$$\sum_{i=1}^{n} M = DDF \sum_{i=1}^{n} T^+ \Delta t , \tag{5}$$

$$DDF = \frac{M_m \cdot \varrho}{T^+} \tag{6}$$

where T+ – sum of positive air temperatures [K] during the same period of n time steps $\Delta t$ [h], DDF – the degree-day factor in mm $d^{-1}K^{-1}$, Mm - measured ablation [m] , M – melting [m w.e.], $\varrho$ – density [kg $m^{-3}$]. Melting is assumed to be zero when the air temperature is $\leq 0°C$.

On the basis of glaciological and meteorological data collected on Werenskioldbreen, daily surface ablation for May - November 2009-2020 was calculated (Figure 7). In the case of gaps in meteorological data collected by the AWS on Werenskioldbreen, data from the Polish Polar Station located 16 km south-east were used (Wawrzyniak and Osuch, 2020). Linear regression was used to fill the gaps ($R^2$ 0.96).

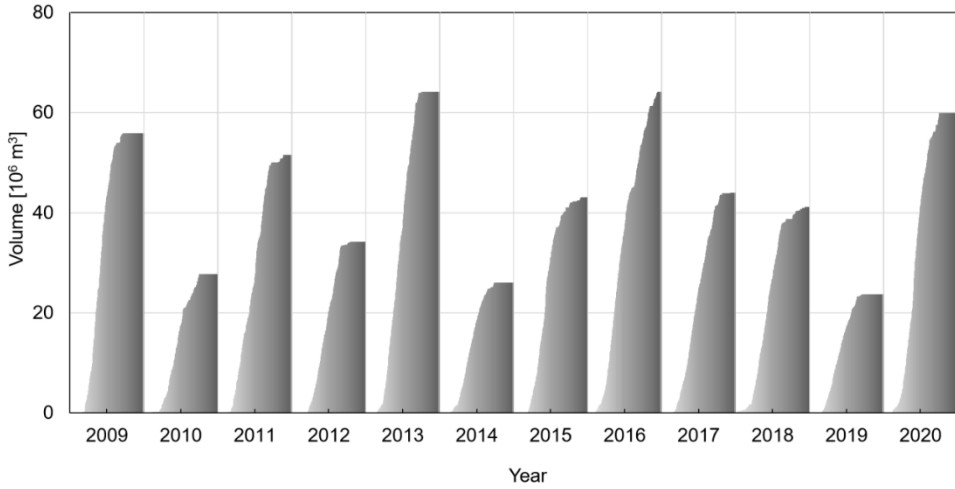

Figure 7: Cumulative ablation [$10^6$ m$^3$] in May – November (2009 – 2020) for Werenskioldbreen.

Seasonal sums of surface ablation oscillate between about $23.7 \pm 1.7$ (2019) and $64.2 \pm 4.5 \cdot 106$ m$^3$ (2013), with an average of $44.7 \pm 3.1 \cdot 106$ m$^3$ for 2009-2020. The value in 2019 may be underestimated due to problems with field measurements caused by the pandemic travel restrictions. The length of the ablation season determines meltwater runoff volumes. It varied in the analysed period from 134 days in 2014 to 203 days in 2016 (the average for 2009-2020 was 163 days). The surface ablation is affected by the decrease in the number of sunny days and the increase of days with precipitation and cloud cover (Wawrzyniak and Osuch, 2020). The amount of water produced by surface ablation is the largest component of the total runoff from the catchment but precipitation can also be an important element of the water balance (Majchrowska et al., 2015).

## 6 Quality control and data processing

Data quality assurance includes additional measurements and calibration of equipment performed during the observation period and post-processing of the collected data. The analysis differed for the meteorological data constituting the time series and for the glaciological data.

The first stage of quality control for meteorological data consisted in visualizing each of the measurement series and reviewing the disrupted data caused by interruptions in the operation of sensors. Due to its location, the automatic measurement station (AWS) operating on Werenskioldbreen could not be maintained with high frequency. As a result, there were periodic problems with the power supply as well as with the freezing of some sensors. Power shortages manifested themselves in the disappearance of measurements and the occurrence of isolated measurements, the correctness of which could not be confirmed, and therefore they were removed. Similarly, malfunctioning of sensors manifested in 'blocking' the measurement at one value for a longer time. It mainly concerned wind speed measurements. As such values are unnatural, they were identified as

erroneous and removed from the set during visual inspection. The next stage of the control was the identification of individual measurements where the values were too different compared with the previous and following measurements and that did not fit in the short-term trend. These data spikess were averaged with respect to adjacent measurements. It mainly concerned air temperature and humidity records, where such spikes are believed to be artefacts. Similarly, the analysis of the measurement series was performed in terms of unnatural values, i.e. values exceeding the permissible variability of the relative humidity or air temperature. These were a few cases. In these situations, such values were eliminated or averaged over adjacent measurements. In the last step, the same variables were compared with records from other weather stations in Svalbard. Air temperature time series have been tested with observations at the Polish Polar Station Hornsund (Wawrzyniak and Osuch, 2020). Mainly, the correlations of the variability of parameters were checked in comparison to the stations accepted as reference. Nevertheless, it should be remembered that even in the case of close points, this correlation does not have to be high or consistent due to the specificity of these stations, i.e. different shading conditions, ground, topography or exposure.

Analysis of collected data revealed some imperfections in 10-minutes measurements of the air humidity. There have been some measurements that slightly exceeded the value of 100%. In this case, one of two procedures was undertaken. If the neighbouring measurements to the questionable record show high air humidity – the exceeding value was reduced to 100%. If the neighbouring measurements to the questionable measurement show low air humidity the value was averaged from these neighbouring measurements.

A separate analysed issue is the variability/fluctuation of measurements in this 10-minute series. A series of unjustified peaks in the measured values were identified, with a relatively small variability of the parameters recorded by the sensors earlier and later. The sites for this potential correction were searched for when reviewing the series on the chart. At the time of identifying such a value, the variability of consecutive measurements was analysed. 12 measurements were analysed before the questioned measurement (± 2 hours). In this situation, when these fluctuations exceeded 2 standard deviations of the variability, they were averaged with the direct measurements before and after the questioned measurement. When more than 3 standard deviations of the measurements assessed according to this criterion were registered, directly after each other, they were completely removed and marked in the published files by missing values (described in the attributes of NetCDF files). This procedure was used for both air humidity and temperature. The tested measurements were compared, if possible, with their counterparts measured at the station in Hornsund, for similar dynamics of variation, which could justify a similar dynamics of variability in the published measurement series. When a large dynamics of the variability of the measured parameters were identified in both locations, the criterion of 3 standard deviations was used instead of the previously described 2SD. Unfortunately, the conditions of the measuring station in Hornsund are very different from the location on the glacier. For this reason, the direct possibility of comparing measurements between these locations may be limited only to the analysis of short-term trends and dynamics of the variability of both air temperature and humidity. The stations differ in height above sea level, distance from the sea, and the ground on which they are located.

In the case of wind measurements, the most common problem was the one resulting from the sensor icing, which manifested itself in recording the same value over a longer period. The only possible correction here was to remove erroneous values throughout the occurrence.

In the case of radiation measurements, the fewest corrections were made as these sensors proved to be the most reliable. The introduced corrections concerned only sporadic jumps in the measured parameters. However, in this case, due to the large impact of cloudiness on the measured parameters, which may be marked in the measurements, only the evident cases were removed or averaged with such fluctuations. Jumps of single measurements against a background of relatively low long-term variability were identified as such cases.

Measurement series prepared and tested in such a way were used to calculate series with an hourly and daily resolution (24h). The series was created as a result of averaging or summing up depending on the parameter under development.

Glaciological data are not collected automatically in large amounts but are based on single, unique observations that must be made with great care as they are not possible to repeat or relate to observations from other areas.

Each measurement of the ablation stake was performed twice. If the funnel melts in ice or snow around the stake, the measurement was made to the theoretical flat surface joining the edges of the funnel. In the event of a stake skewing, its total length was measured and then, if possible, the stake was replaced with a new one. Measurements of the snow depth, apart from making snow pits or shallow drilling, were always verified by taking 2-3 measurements with an avalanche probe. In order to obtain comparable measurements of bulk snow density (and SWE) these measurements were performed with two different methods (snow pit and shallow drilling), and a series of parallel measurements were performed showing that the difference in the calculated SWE does not exceed 5%. In order to obtain the most accurate data from the ICS drill, the quality of the obtained ice and snow cores was checked in order to determine the precise diameter of the obtained cores.

The obtained point and glacier-wide surface mass balance calculations were compared with the data published by the World Glacier Monitoring Service (https://wgms.ch) for other glaciers on Svalbard in order to verify the consistency of trends (Schuler et al., 2020). Data on surface ablation in seasons where it was possible were controlled by comparison with the data collected by the SR50 (sonic ranger) sensor, which was also used to verify the duration of the ablation season. The glaciological data were saved in the CSV files.

The quality of DEM generated from the SPOT images in 2008 was validated with the height of stakes on Werenskioldbreen (Ignatiuk et al., 2014). The median value and standard deviation of the accuracy of the DEM were -0,85 m and 2.2 m, respectively. Validation of the DEM generated from Pleiades images taken in 2017 was based on stake positions over neighbouring Hansbreen (Błaszczyk et al. 2019). The median value and standard deviation of DEM accuracy were -0,36 m and 0.24 m, respectively.

## 7. Dataset structure

Prepared measurement series were saved in the NetCDF (Network Common Data Form) format and placed on the server supporting OPeNDAP (www.ppdb.us.edu.pl). The choice of this type of file is due to its universal nature. NetCDF files are in line with the modern trend of storing and publishing measurement series meeting the FAIR data principles. The collections are compliant with Unidata's Attribute Convention for Dataset Discovery (ACDD-1.3) and Climate and Forecast (CF) Conventions (CF-1.8). The Attribute Convention for Dataset Discovery identify and define a list of NetCDF global attributes recommended for describing a NetCDF dataset to discovery systems such as Digital Libraries. Software tools can use these attributes for extracting metadata from datasets, and exporting to Dublin Core, DIF, ADN, FGDC, ISO 19115 etc. metadata formats. The CF metadata conventions are designed to promote the processing and sharing files created with the NetCDF API. The conventions define metadata that provide a definitive description of what the data in each variable represents and the spatial and temporal properties of the data. This enables users of data from different sources to decide which quantities are comparable and facilitates building applications with powerful extraction, regridding, and display capabilities. The CF convention includes a standard name table, which defines strings that identify physical quantities. Global Attributes of prepared NetCDF files comply with the recommendations of The Arctic Data Center (ADC) which is a service provided by the Norwegian Meteorological Institute (MET) (https://adc.met.no/node/4).

All ACDD 1.3 Variable Attributes recommended were used. They were supplemented with the so-called _FillValue = -999.9 indicating data gaps and valid_max and valid_min describing the natural and allowed variability of these parameters in the measurement area. All measurement parameter names follow Climate and Forecast (CF) Standard Name Table version 77 which was available on the day when the dataset was published.

The keywords vocabulary used is consistent with the Global Change Master Directory (GCMD) Keywords (https://earthdata.nasa.gov/earth-observation-data/find-data/idn) developed for 20 years by The National Aeronautics and Space Administration (NASA)/gcmd-keywords) which are a hierarchical set of controlled Earth Science vocabularies that help ensure Earth science data, services, and variables are described in a consistent and comprehensive manner and allow for the precise searching of metadata and subsequent retrieval of data, services, and variables.

## 8. Data availability

The data is stored in two repositories that provide long-term availability, open access, DOI and license according to the FAIR principles: Zenodo (www.zenodo.org):

meteorological data: https://doi.org/10.5281/zenodo.6528321 (Ignatiuk, 2021a), glaciological data: https://doi.org/10.5281/zenodo.5792168 (Ignatiuk, 2021b). During the review of the article, successive versions of the data were corrected and updated (versions 1-4). The final version of the datasets for meteorological data is version 4 and for glaciological data is version 1.

and Polish Polar Database (https://ppdb.us.edu.pl/):

Air temperature:

https://ppdb.us.edu.pl/geonetwork/srv/pol/catalog.search;jsessionid=7A0C3C8EAEA1B8F61D8F0B57177B7098#/metadata/abc6becf-97f0-4dca-b597-2fa3438f43ab

Relative humidity:

https://ppdb.us.edu.pl/geonetwork/srv/pol/catalog.search;jsessionid=7A0C3C8EAEA1B8F61D8F0B57177B7098#/metadata/bdd6b724-d75c-49a1-83c6-eb2007107cde

Wind speed:

https://ppdb.us.edu.pl/geonetwork/srv/pol/catalog.search;jsessionid=7A0C3C8EAEA1B8F61D8F0B57177B7098#/metadata/d0ad64ab-ad70-43d7-9383-8a9213e6c40f

Shortwave flux:

https://ppdb.us.edu.pl/geonetwork/srv/pol/catalog.search;jsessionid=7A0C3C8EAEA1B8F61D8F0B57177B7098#/metadata/12ed9717-8cd7-4583-b2c6-089d50e6ad61

https://ppdb.us.edu.pl/geonetwork/srv/pol/catalog.search;jsessionid=7A0C3C8EAEA1B8F61D8F0B57177B7098#/metadata/fa3bd41b-dfbb-49e8-bdf6-7c56e9bb902f

Longwave flux:

https://ppdb.us.edu.pl/geonetwork/srv/pol/catalog.search;jsessionid=7A0C3C8EAEA1B8F61D8F0B57177B7098#/metadata/9309a6b1-663c-4227-9eb6-39761c1d868d

https://ppdb.us.edu.pl/geonetwork/srv/pol/catalog.search;jsessionid=7A0C3C8EAEA1B8F61D8F0B57177B7098#/metadata/5aa3b739-af33-4e57-bf68-7a8757985b2d

In addition, the glacier mass balance data are stored in the World Glacier Monitoring Service database (dx.doi.org/10.5904/wgms-fog-2021-05, WGMS, 2021) and INTAROS Data Catalogue (https://catalog-intaros.nersc.no/dataset/glacier-mass-balance-werenskioldbreen).

All the data are also available through the Svalbard Integrated Arctic Earth Observing System (SIOS) data access portal
(https://sios-svalbard.org/metsis/search).

**9 Summary**

This paper has presented details of the glaciological and meteorological dataset (2009-2020) from the Werenskioldbreen (Svalbard). The meteorological dataset includes 10 minutes, hourly and daily air temperature, relative humidity, short- and long-wave radiation, and wind speed. The glaciological dataset includes point surface mass balance (winter, summer, net),
snow depth, bulk density, and snow water equivalent (SWE) for the mass-balance stakes, annual glacier-wide surface mass

balance and modelled daily surface ablation. These data allow observations of the rapid changes taking place in the Arctic. In particular, they allow determining the rate of climate change directly on glaciers. Werenskioldbreen mass loss is accelerating at a rate of -0.23 m w.e. for a decade. These observation data have been already used to assess the hydrological models and glaciological studies. The objective of releasing these data is to improve the usage of this data to calibration and validation of the remote sensing products, models as well as to increase data reuse (Moholdt et al., 2010; Möller et al., 2011; Claremar et al., 2012; Østby et al., 2014; Błaszczyk et al., 2019).

## Acknowledgements

The study presents part of the results from the project "Hindcasting and projections of hydro-climatic conditions of Southern Spitsbergen" (grant no. 2017/27/B/ST10/01269) financed by the Polish National Science Centre and the "Arctic climate system study of ocean, sea ice, and glaciers interactions in Svalbard area"—AWAKE2 (Pol-Nor/198675/ 17/2013), supported by the National Centre for Research and Development within the Polish–Norwegian Research Cooperation Programme and the SvalGlac—Sensitivity of Svalbard glaciers to climate change, the ESF Project. Glaciological and meteorological data have been processed under assessment of the University of Silesia in Katowice data repository within the project Integrated Arctic Observing System (INTAROS). This project has received funding from the European Union's Horizon 2020 research and innovation programme under grant agreement No. 727890. A legacy from the ice2sea 7th FP and the ESF SvalGlac projects was used. The studies were carried out as part of the scientific activity of the Centre for Polar Studies (University of Silesia in Katowice) with the use of research and logistic equipment (monitoring and measuring equipment, sensors, AWS'es, GNSS receivers, snowmobiles and other supporting equipment) of the Polar Laboratory of the University of Silesia in Katowice
The authors would like to thank the employees, doctoral students and students of the University of Silesia in Katowice for their help in the fieldwork. We would also like to thank our colleagues from the University of Wrocław for their hospitality at the Polar Station. Stanisław Baranowski Spitsbergen and excellent cooperation. Also thanks to the members of the expedition from the Polish Polar Station Hornsund for their cooperation. Thanks to Inger Jennings for the linguistic proofreading.

## Competing interests. a

The contact author has declared that neither they nor their co-authors have any competing interests

## Author Contribution

DI- Conceptualization, Formal analysis, Methodology, Validation, Writing – original draft preparation, review & editing. MB - Investigation, Methodology, Writing – review & editing. TB - Data curation, Investigation, Writing – review & editing. MG - Investigation, Methodology, Validation, Writing – review & editing. JJ – Conceptualization, Funding acquisition,

Supervision, Writing – review & editing. MK - Formal analysis, Investigation, Writing – original draft preparation. ML - Investigation, Validation, Writing – review & editing. ŁM - Data curation, Formal analysis, Writing – original draft preparation, review & editing. ŁS - Investigation, Writing – review & editing.

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
