# Peer review of "A decade of glaciological and meteorological observations in the Arctic (Werenskioldbreen, Svalbard)"

_Earth System Science Data, 2021_

## Author Response (AR1)

Dear Editor and Reviewers,

We would like to kindly thank you for your time and constructive comments. We have carefully studied all remarks. We made appropriate corrections in the manuscript and prepare the answers for the Reviewers.

Below you will find all of our answers to your comments.

**Answer to Referee 1 comments:**

**RC1**: 'Comment on essd-2021-464', Anonymous Referee #1, 20 Jan 2022

*This paper quickly present the main results of about 10 year of meteorological and glaciological measurements taken at an arctic site in Svalbard. The contribution is certainly of high value in terms of data provision and I suggest its pubblication after few minor corrections, see below.*

The main issue I see, maybe not strictly related to the paper, is the fact that following the link https://ppdb.us.edu.pl it's not easy to find the data pertaining to this paper. It's not immediately clear how to reach Geonetwork or OPeNDAP HYRAX and the datasets descripted in Geonetwork don't contain the direct links to the data files. I would suggest to include the links in it. It was easier to follow the Zenodo links.

**Answer: Thank you for paying attention to this. There were some problems connecting HYRAX and THREEDS to Geonetwork (especially for Chrome). Previously, the links were for access via OPeNDAP. Most of the problems have been resolved and direct links to datasets in Polish Polar Database have been added to the data availability section**.

Minor comments:

L32-33 please provide a reference for the climatological normal. If the references are the links reported at L35, I would suggest to declare them earlier in the text. Something like "according to…"

L41 to which specific "disparities" are you referring to? Please clarify.

L47 "is a well-studied…"

L49 "glacier covered an area…" I suggest to use the past form.

L50 I would suggest to specify that Slyngfjellbreen is the south-east part of Werenskioldbreen.

L73 "downwelling" instead of "upwelling"?

Table1 Accuracy of the HMP155 sensor is not correct.

**Collective answer to the above comments: Corrected.**

Table1 HMP45AC is from HOBO?

**Answer:** No, it is Vaisala sensor. Corrected.

L91 Specify the month(s) corresponding to the end of the ablation season, as done for the accumulation season.

L96 boreholes have been drilled "on" the stakes? You mean "at" or "near"?

L111 SWE already defined at L93

L116 "and another from Pleiades…"

L131 did you checked the meaningfulness of these trends?

**Collective answer to the above comments: We have calculated the significance of the trend presented in this study using the non-parametric modified Mann – Kendall test (Hamed and Rao, 1998) with considering the effect of autocorrelation of time series. The slope of the trend was calculated using Sen's method (Sen, 1968). The test indicated the statistically significant increasing temperature trend in the period 2010-2016 (with the significance level alfa = 0.05 and p-value = 0.036) taking into account the 12-month seasonality the Sen's slope was 0.02. The text in the manuscript has been corrected.**

L135 I would say that the height of the sun is the first responsible for daily variations.

**Answer: Agree. Corrected.**

L140 Could you provide estimates of albedo for snow and ice obtained from your measurements?

**Answer: Based on the collected data on the upwelling and downwelling of short-wave radiation, calculating the albedo for ice and snow does not pose major problems. Nevertheless, the issue of albedo, its variability, and modelling is the subject of another paper which the authors work on, which is based on the data presented in this manuscript. So we are not going to present the results of albedo in this paper.**

L155-163 I would like to see here some description on how the point mass balance is calculated. There is only a light sentence like "based on physical parameters of snow and ice".

**Answer: The description has been extended.**

Figure3 Please correct the y label "equivalent". Also, I would draw the zero line.

**Answer: Corrected**

Figure4 Whiskers are difficult to see. And please try to explain better what they represent. What does it mean "an error"?

**Answer: The whiskers present the error estimated using the total differential function. The size of the measurement errors is smaller than the size of the points on the plot that's why it is hardly visible on the graph.**

L222 I'm not sure is correct to use the expression "increases by -014". I would use "decrease by 0.14" also for the summer balance.

Eq 2 and 3 What is n? What is deltat? I would like to see a clearer description of the equations.

**Answer: The description has been extended.**

L245 Melting is assumed to be zero when the temperature is positive?

**Answer: Corrected.**

**Answer to Referee 2 comments:**

**RC2**: ['review report essd-2021-464'](), Anonymous Referee #2, 24 Feb 2022

*Summary:*

*A data-paper serves usually a dual purpose and is somewhere between a technical report describing the measurements/ dataset and a research paper presenting detailed analyses. The manuscript addresses both, and describes datasets from Werenskioldbreen, a glacier in the Arctic archipelago of Svalbard, that cover both glacier mass balance measurements and operation of an automated weather station. These data have been made available and the manuscript describes the meteorological and glaciological measurements, presents some characteristics of each dataset and puts them in relation to each other.*

*I applaud the authors for collecting and publishing these datasets from a generally data-sparse environment such as the Arctic and I will recommend publication of the manuscript once some of the identified shortcomings have been rectified.*

- *Description and presentation of data: although nominal sensor uncertainties are stated in Tab 1, I miss a more thorough discussion of data quality, clearly stated quality thresholds used for discarding data, procedures for gap filling etc. The problem of varying sensor hight above the surface is not at all addressed. Not much is learned from e.g. Fig 2 that presents 7 years of data in hourly resolution. Instead, values could be presented aggregated in monthly boxes to enhance figure readability (for instance Schuler et al. 2014). Adding supporting information about data completeness/ quality would give the reader a quicker overview than a listing of all gaps in the main text. Some figures present uncertainties, but it is never explained what these uncertainties represent (standard deviation, RMS error or other measures?).*

- *Datasets in the zenodo repository come in plain CSV format without explanatory readme-file or other metadata. Even with the manuscript at hand, it needs some intuition to identify the individual variables and which units the values are affiliated with. This must clearly improved to make the dataset useful. At the other repository https://ppdb.us.edu.pl, the meteorological data are available in netCDF format and contain all required metadata. This should be the case also for the data on zenodo. Especially given the difficulties finding the data at PPDB, that the other reviewer already commented on.*

- *Throughout the manuscript, the authors repeat statements about the significance of the dataset, but after all these statements appear more persuasive rather than convincing and would benefit from better support (or more modest wording).*

- *Related to the point above, the analyses to show the significance of the data should be improved; 10 years too short for climate trend analysis. Instead the data could be discussed in light of existing longterm series from Hornsund. Do trends agree? How do differences vary (seasonally, decade)? Such a discussion would provide good arguments for the necessity of these measurements, why should we need additional series if they are only linearly related to other operational series?*

**Answer: The methods of calculating errors were added. The chapter about the data preparation, quality thresholds and procedures for gap filling were refined. We've added also the information**

and analysis about the changes in the height of the sensors above the glacier surface. Thank you for your comment and for drawing attention to the technical shortcomings of the data repositories. Data records in zenodo have been improved. They are in analogues format both for the PPDB and in NetCDF. The direct links to datasets in Polish Polar Database have been added to the data availability section. The importance of such data as those presented in the article is indicated, among others Moholdt et al., 2010; Claremar et al., 2012; Möller et al., 2011; Østby et al., 2014; Błaszczyk et al., 2019). Appropriate references have been added to the text of the manuscript.

Detailed answers are provided under the reviewer's comments below.

Details:

L12: unique…in what sense?

**Answer: Each data from the glacier is unique for a glaciologist :) The wording has been corrected on 'important'.**

L15: 10 minutes, hourly and daily resolution

L15: snow data is not from 49 individual sampling locations but the dataset consists of 49 datapoints, reword the sentence.

L16: spell out SWE (undefined abbreviation).

L16: since you refer to Cogley et al 2011, you should stick with its terminology: …consists of seasonal and annual surface mass balance measurements… ("surface annual winter, summer and net balance" is at least confusing if not inappropriate).

**Collective answer to the above comments**: Corrected.

L18: this is an example for the unsupported, persuasive wording mentioned above: "high-quality and long-term datasets".the quality is not well assessed in the manuscripts and 10 years is not really long-term.

**Answer: Agree. Corrected.**

L18: …to serve as local forcing data…

L24: "…and significantly expand our knowledge" not per se, but they may contribute to that. Such a bold statement would require a reference.

**Collective answer to the above comments: Corrected.**

L26: "…does not adequately address…" why not? needs a reference

**Answer: The significance of direct measurements was pointed by Pelliccotti et al., 2014; Gabbi et al., 2017. The text has been corrected.**

L27: Both, atmosphere and ocean… (both are parts of the climate system).

L28: important references addressing climate change in Svalbard: Vikhamar-Schuler et al 2016, Hanssen-Bauer et al 2019, Førland et al 2020.

**Answer: Added**

L31: define JJA

L33: ...the highest air temperature since beginning of the measurements... (add a reference)

**Answer: Corrected. Own study based on data from the SIOS and Hornsund databases.**

L37: minimum extent in the history...(add reference)

**Answer: Added**

L40: add a reference

L40/41: what disparities? I do not understand.

**Answer: Corrected to 'transition'.**

L43: must be...developed in Svalbard. Why there?

**Answer: Corrected: "Therefore, monitoring of such parameters and the environmental response to climate change is recommended to be carried out in Svalbard, where climate warming is one of the most dynamic (Nordli et al., 2014; Isaksen et al., 2016)".**

L47: ...located in South Spitsbergen (Fig. 1)...

L49 ...covered ...in 2008...

L51: what is "the internal marginal zone"?

**Answer: Corrected to 'glacier's forefiled'**

L53: ...to be treated as a well-defined ...

L55: please mark the location of the Baranowski station on the map (Fig 1)?

**Answer: Added**

L66: "...placed in the ice (drilled ca 6 m deep)" why not using a floating station in the ablation area?

**Answer: During the installation of the station in 2009, it was assumed that it was in the accumulation zone. Climate warming and local conditions have verified this assumption. We are currently preparing to set up a modern floating station in this area.**

L80: between the glacier surface and sensor level

L80: ...were systematically replaced... how often? Every year? More often? Or only if needed?

**Answer: Damaged sensors were replaced as soon as a failure was noticed - usually during spring or autumn service.**

Tab 1: I understand that sensor level above surface will continuously vary, but at least you could indicate original height or the target heights at which you wish the sensors to be...

**Answer: Added.**

L83: is this the measurement interval or the recoding interval?

**Answer: measurement interval. It has been corrected in the text.**

L87: The locations have been chosen to cover the elevation range... (combine with next sentence)

L90: did you really use an ice core drill or the Kovacs stake auger? (same brand, different tool)

**Answer: Yes we use ice core drill. We didn't have the Kovacs stake auger in our inventory at the time.**

L92: the properties of the snowpack

L101: …renovated during maintenance visits.

L102: …pandemic travel restrictions inhibiting access to field area. (please state the years when this happened)

Tab2: please identify coordinates with Northing and Easting. The coordinates of stake #9 apparently have been swapped.

What is the value of labeling GPS and SR50 measurements if these are not presented?

L110: …annual and seasonal point mass balance. (you do not measure ablation or accumulation but mass balance!)

L111: what you refer to as "surface mass balance" seems to correspond to the "glacier wide mass balance". The first considers only mass changes at the surface of the glacier, the second is the integral over the surface area of the glacier, see Cogley 2011.

**Collective answer to the above comments: Corrected**

L155ff: which DEM did you use for which period? How did you deal with transitions between them?

**Answer: We've used 2008 DEM for the calculations for 2008-2019 period and the new 2017 DEM for the latest 2020 WGMS report. We did not have access to this product before. There were no special transitions between these DEMs. Currently, we have collected more DEMs for the period 2008-2020 and earlier, on the basis of which we are going to prepare a data correction for WGMS and an article on the recalculation of the mass balance of Werenskioldbreen and Hansbreen. The text in the manuscript has been supplemented with relevant information.**

L120: travel restrictions started in Mar 2020, so they cannot be blamed for data gaps in 2017-2020…

Fig2: how much are the values influenced by changing sensor levels as the surface melts down/ sensors are relevelled? Add some information about completeness of the records (e.g. percentage of time). The shortwave radiation records could be analyzed in terms of albedo and its seasonal changes; adding potential clear sky radiation would give an impression about cloud effects.

For a melting snow/ glacier surface, the surface temperature is capped at the melting point and so is the outgoing longwave radiation (316 W m-2). Discuss why you observe upwelling radiation in excess of this value!

**Answer: Based on the collected data on the upwelling and downwelling of short-wave radiation, calculating the albedo for ice and snow does not pose major problems. Nevertheless, the albedo, its variability, and modelling of the albedo is the subject of another scientific paper which the authors are working on, that's way we do not want to present the results of albedo in this paper.**

**Increasing distance between the glacier surface and the sensors during the season is affecting on the air temperature measurements. Periodic measurements of the vertical temperature gradient between 0.5 and 4 m carried out at the AWS indicate that the air temperature in the atmospheric boundary layer changes by about 0.2oC per 1 m during the ablation season. Assuming the**

homogeneity of the surface around AWS, increasing the distance of the CNR4 sensor from the glacier surface should not affects its measurements,.

  The values above 316 W m-2 of outgoing longwave radiation may be caused by the presence of the water under the station or the presence of sediment or cryoconite. Both of these situation were observed at AWS. In the he view of the CNR4 sensor (180 °), there is also a mast with a logger, sensors and a solar panel, what cause distorts of the observations. These problems are difficult to eliminate.

Information about the completeness of datasets has been added to Table 2 and the text in the manuscript has been supplemented with relevant information.

L130 …the difference …between…and…

L131: how does the temperature lapse rate vary with time?

**Answer: 0.55-0.80 Added to the text**

L131: deriving a temperature trend from a 7 years series is questionable, especially without further remarks on statistical significance. How does this compare to the long-term series from nearby Hornsund?

L133/134: how much is the trend in summer influenced by the increasing distance between the sensor and the cooling surface as the snow/ ice melts down?

**Answer: We have calculated the significance of the trend presented in this study using the non-parametric modified Mann – Kendall test (Hamed and Rao, 1998) with considering the effect of autocorrelation of time series. The slope of the trend was calculated using Sen's method (Sen, 1968). The test indicated the statistically significant increasing temperature trend in the period 2010-2016 (with the significance level alfa = 0.05 and p-value = 0.036) taking into account the 12-month seasonality the Sen's slope was 0.02. Gaps in the data were filled based on the relations between air temperature measured on the PPS in Hornsund and air temperature on the WRN glacier (R2 = 0.96). Furthermore, the issue of changing the height of the sensors at the glacier is also not negligible here - the potential values of the vertical gradient have been discussed above.**

L145: In 2009-2020, the AWS measured… this is not continuously measured at all: rel humidity was recorded in 2011 and then again 2016-2019, wind direction only since 2016 (according to information given in Tab1).

**Answer: Corrected.**

L148: if the sensors are not robust enough for polar conditions why did you not use others that are better suited?

**Answer: We use high-quality sensors. However, in stations that are left on the glacier for long periods without servicing, they fail more quickly. The sentence has been corrected.**

L150: …are of great value for solving specific scientific problems. Could you give some examples to support this claim?

**Answer: Added**

L161: no, you cannot measure these by only 1-2 visits, you do measure the balance!

**Answer: Agree, the text has been corrected.**

L161: …on the physical properties…

**Answer: The sentence has been changed.**

L163: better description of error estimation is required.

**Answer: Added**

Fig 3: y-label should be " Water equivalent"

Why are the stake values not sorted according to stake number? This is a confusing sequence…

**Answer: The stake values are sorted according to the height above sea level. The description has been added in the text.**

L165ff: the observation that the variability of annual balances is dominated by the variability of summer balances is not new, add references. This is actually much better seen in Fig 6.

**Answer: Added**

L178: …was generally low.

L181: …for calculations of the glacier-wide surface mass balance.

L188: …was then averaged weighted by layer thickness.

L189: mention more about the method of Sturm (2010)

**Answer: Corrected**

Tab3: add the SWE values as well.

**Answer: Added**

How did the snow density vary in space/ with elevation?

**Answer: There no correlations between snow density and the snow cover depth or elevation. This was also described in the manuscript.**

Fig 5: explain which error measure is indicated by the whiskers.

Present the coefficients of the regressions shown in both plots. The slope of the line in Fig 5a would correspond to the bulk density. How much of the variability in Fig5b is due to variability in snow thickness versus variations in density? Better to present the measurements instead of the inferred quantity.

**Answer: The coefficients of the regression and error equations have been added. The snow depth is very highly and positively correlated with the elevation (correlation coefficient was 0.85 for $p < 0.05$)), but there was no correlation between the bulk snow density with the elevation (-0.07) and snow depth (0.16). Base on the regression analysis of the SWE and the single parameter - snow density ($\rho$), elevation and snow depth (hs), explains 13%, 62%, 96% of the SWE respectively. The variability in Fig5b. is dependent more on the snow depth than snow density. The list of average SWE values has been added to Table 2.**

L199: $R^2 = 0.62$ is not "very high correlation" !!

L206: as mentioned above, your "surface mass balance" corresponds to "glacier-wide mass balance" (Cogley 2011)

Fig 6: typo in label on y-axis: Surface mass balance

**Collective answer to the above comments: Corrected**

State the trends and add ameasure for their significance.

**Answer: The equations of the trends were added.**

**We have check the significance of the trends using the non-parametric modified Mann – Kendall test (Hamed and Rao, 1998). The slope of the trend was calculated using Sen's method (Sen, 1968). The test with the significance level alfa = 0.05 indicated that there is no statistically significant trend in 2009-2020. The Sen's slope was -0.01, -0.12 and -0.02 for winter, summer and annual glacier-wide mass balance respectively.**

Comment on the records winter balance in 2014. Has this also been observed on other glaciers?

**Answer: Thank you for very good point. Indeed, there was an error in the calculation of glacier-wide winter mass balance in 2014. The Bw for 2014 was corrected and all other mass balance calculations were checked. There are no other errors.**

L 216 fff: again: summer and winter mass balances

L222: summer balance decreases by -0.14 m (negative value marks a decrease!!)

L222: …acceleration of mass loss.

L226: …according to the model…

L228: the summer balance decreased from -1.23…to -1.68

**Collective answer to the above comments: Corrected**

L230: observations (1999-2019)…this is confusing, you present data for 2009-2019

In the following lines you fuse the model results and your measurements to infer a long-term evolution. This is questionable as long as you do not show the level of agreement between model and observations. However, this is not possible to derive since the two series do not overlap in time (model 1959-2002, measurements 2009-2019)

**Answer: Grabiec et al. (2012) compared the modelling results with the observations. The average correlation coefficient of the modelled and observed mass balance in the 5 seasons (1994, 1999–2002) was 0.67 (meteorological data) and 0.70 (ERA-40). Additional information and corrections have been added to the text of the manuscript.**

L247: better explain how the error estimate has been derived

Answer: Added

L248: better explain the gap filling: has the Hornsund data been adjusted using a regression or similar?

**Answer: Description has been added in the chapter on meteorological data.**

Fig 7, caption is misleading, the figure shows cumulative ablation, not daily ablation.

What value for DDF has been used? How has it been determined/ selected?

How are eqs 2 and 3 used in the calculations? For the entire glacier? Or per stake location (and according adjustment of temperature records)?

**Answer: For each season, a separate DDF was determined based on air temperature data and cumulative ablation calculated from glacier-wide mass balance. Equations 2 and 3 were used for the entire surface of the glacier.**

L254: ….determines meltwater runoff volumes.

L265: …could not be maintained with high frequency.

L268: Similarly, malfunctioning of sensors…

L272: These data spikes…

L273: …such spikes are believed to be artefacts.

L274: …exceeding a permitted variability… (state the value of this important threshold!)

L276: …compared with records from other weather stations…

**Collective answer to the above comments**: Corrected

Could you show or discuss the results of these comparisons? Show the agreement or at least state a quality measure.

are replaced/ corrected values in the published datasets marked as such? (distinct from valid measurements)

**Answer: Additional description of the analysis and comparison of the data has been added to the manuscript (LXXX-YYY). The data in the database does not contain any completed gaps made with the use of data from Hornsund. Comparing such data with the data from Hornsund would be unjustified.**

L284-306 do not belong to a quality-control section but rather to a separate section "dataset structure"

**Answer: New section has been added**

L307: the first sentence can be removed without losses.

L310-317: consider moving this to the "methods" section

**Answer: After adding a new section "Dataset structure" this paragraph fits here.**

L318: what are the results of this comparison?

**Answer: References have been added. Further analyzes will be presented in the forthcoming article on the recalculation of the mass balance of Werenskioldbreen and Hansbreen.**

L321-326: consider moving this to the "methods" section

L348: "high-quality and long-term" …see comments above.

**Collective answer to the above comments: Corrected**

**Answer to Editor comments:**

1. You state that the files are CF and ACDD compliant. I ran one through 'cchecker.py' (the IOOS compliance checker) and there are several ACDD and CF issues. Please run a CF-compliance checker on your NetCDF files. Please use ISO 8061 date formats (YYYY-MM-DD).

**ANSWER: Thanks for your comments. NetCDF files have been supplemented with additional information, the dates to ISO 8061 standard have been corrected and the problem with ACDD has been solved.**

2. Line 32: I do not think "Svalbard airport" is the correct name.

**ANSWER: The station is situated at the airport of Svalbard, which in Norwegian is called Svalbard lufthavn, and this is the name used in Norwegian station lists.  In English-language publications, the name Svalbard Airport is used.**

**Nordli, Øyvind. (2010). The Svalbard Airport Temperature Series. Bulletin of Geography, Physical Geography Series. 3. 5-25. 10.2478/bgeo-2010-0001.**

3. Table 2: Coordinates have mm resolution?

**ANSWER:  It has been corrected.**

4. Line 245: Melting is assumed to be zero when air temperature is >= 0 °C ? Should this be <= 0?

**ANSWER: This is a mistake. It has been corrected.**

We hope that the changes we have made will be satisfying.

Sincerely,

Authors

---

## Author Response (AR2)

Dear Editor,

NetCDF files have been corrected and uploaded as version 3 to Zenodo and PPDB.

All errors have been corrected. The report from the IOOS Compliance Checker Report can be found below.

Best regards,

Authors
* * *
IOOS Compliance Checker Report

Version 5.0.0

Report generated 2022-04-22T11:45:40Z

acdd:1.1

http://wiki.esipfed.org/index.php?title=Category:Attribute_Conventions_Dataset_Discovery
* * *
All tests passed!

---

## Author Response (AR3)

Dear Editor,

Thank you for checking the article in detail and for your comments.

Data version information has been added to the manuscript. The DOI has also been updated.

The NetCDF files have been corrected and the dataset has been upgraded to version 4.

Best regards,

Authors